

# Nitrogen fertilization affects maize grain yield through regulating nitrogen uptake, radiation and water use efficiency, photosynthesis and root distribution

Wennan Su[1,2,3], Shakeel Ahmad[1,2], Irshad Ahmad[1,2] and Qingfang Han[1,2,3]

[1] Key Laboratory of Agricultural Soil and Water Engineering in Arid and Semi-arid Areas, Ministry of Education/Institute of Water Saving Agriculture in Arid Areas of China, Northwest Agriculture and Forestry University, Yangling, China
[2] Key Laboratory of Crop Physio-ecology and Tillage Science in North-western Loess Plateau, Ministry of Agriculture/College of Agronomy, Northwest Agriculture and Forestry University, Yangling, China
[3] College of Agronomy, Northwest Agriculture and Forestry University, Yangling, China

Corresponding author
Qingfang Han,
hanqf88@nwafu.edu.com

## ABSTRACT

High external nitrogen (N) inputs can maximize maize yield but can cause a subsequent reduction in N use efficiency (NUE). Thus, it is necessary to identify the minimum effective N fertilizer input that does not affect maize grain yield (GY) and to investigate the photosynthetic and root system consequences of this optimal dose. We conducted a 4-year field experiment from 2014 to 2017 with four N application rates: 300 ($N_{300}$), 225 ($N_{225}$), 150 ($N_{150}$), and 0 Kg ha$^{-1}$ ($N_0$) in the Northwest of China. GY was assessed by measuring the photosynthetic capacity and root system (root volume, surface area, length density and distribution). Grain yield decreased by −3%, 7.7%, and 21.9% when the N application rates decreased by 25%, 50%, and 100% from 300 Kg ha$^{-1}$. We found that yield reduction driven by N reduction was primarily due to decreased radiation use efficiency (RUE) and WUE instead of intercepted photosynthetically active radiation and evapotranspiration. In the $N_{225}$ treatment, GY, WUE, and RUE were not significantly reduced, or in some cases, were greater than those of the $N_{300}$ treatment. This pattern was also observed with relevant photosynthetic and root attributes (i.e., high net photosynthetic rate, stomatal conductance, and root weight, as well as deep root distribution). Our results suggest that application of N at 225 Kg ha$^{-1}$ can increased yield by improving the RUE, WUE, and NUE in semi-arid regions.

## INTRODUCTION

In the past four decades, global maize production has greatly increased (*FAO, 2018*) mainly due to application of nitrogen (N) fertilizers. Worldwide, N fertilizer has widely been excessively applied to achieve higher grain yields (*Cui et al., 2009*; *Meng et al., 2016*; *Liang et al., 2020*). For example, the average dose of N fertilizer applied by the farmers was

greater than 300 Kg ha$^{-1}$ (288 ± 113 Kg ha$^{-1}$), which far exceeds the optimal N rates for maize demonstrated in field experiments (*Zhang et al., 2015a*; *Chang et al., 2014*; *Yang et al., 2017*). N fertilizer was applied in excess (350–600 Kg ha$^{-1}$ year$^{-1}$, *Mueller et al., 2013*) in an attempt to maximize yields in the North China Plain. However, excessive application of N fertilizer has negative effects on crops, greatly reduces N use efficiency (NUE), and causes significant nitrate leaching losses (more than 50% N to the environment) and contamination of groundwater (*Erisman et al., 2013*; *Wang et al., 2014*, *2019a*; *McBratney & Field, 2015*; *Ahmad et al., 2018*; *Suchy et al., 2018*). Reducing N input rates from this level to "moderate" levels in maize fields may improve NUE, maintain a fair level of maize grain yield (*Zhao et al., 2010*; *Dai et al., 2015*, *Qiang et al., 2019*), and display less negative environmental impacts. In order to implement reduced N input rates, it is necessary to assess the extent to which the N fertilizer application rate is consistent with crop N requirements to maximize resource utilization and maintain relatively high grain yields (*Robertson & Vitousek, 2009*; *Zhang et al., 2015b*).

Radiation interception and radiation use efficiency (RUE) form the basic framework for analyzing crop yield constraints. Variations in crop biomass due to abiotic factors may be attributed to intercepted photosynthetically active radiation (IPAR), RUE, or the combination of both IPAR and RUE (*Fletcher et al., 2013*). Reduced leaf area under low-N conditions is accompanied by a reduction in radiation interception (*Massignam et al., 2012*). Low-N conditions primarily decrease the photosynthetic rate per unit area (*Vos, Van Der Putten & Birch, 2005*), indicating that a low N effect on both leaf size and photosynthetic capacity may affect the final grain yield. Understanding how maize production and resource use are affected by varying N application rates will inform improvement of N fertilizer management to achieve optimal grain yield and resource use efficiency. The appropriate amount of N fertilizer input can improve utilization of precipitation during the crop season (*Dai et al., 2015*; *Herrera et al., 2016*; *Yang et al., 2017*; *Qiang et al., 2019*). Previous studies of the effect of N fertigation on grain yield and resource use efficiency primarily describe the effects of single resource utilization at the leaf or plant level (*Brown, Jamieson & Moot, 2012*). A preceding study described the relationship between maize water use efficiency (WUE) and NUE in pot conditions (*Wang et al., 2019b*). However, few comprehensive studies on the effects of N fertilizer on the utilization of radiation, water, and N resources in field crops have been performed.

Enhancing photosynthesis is critical to promoting crop yield (*Mu et al., 2017*). In order to understand the underlying mechanisms and differences in photosynthetic capacity, it is necessary to determine the relevant photosynthetic parameters of ear leaves (*Li et al., 2013*; *Zhang et al., 2017*; *Lamptey et al., 2017*). Root morphology and distribution also play key roles in the acquisition of soil resources such as nutrients and water (*Ristova & Busch, 2014*; *Lynch, 2013*; *Yu et al., 2014*). *Mi et al. (2010)* proposed the ideotype root architecture of "high yield and high N efficiency" in maize, which provided references for root research. In field conditions, the complexity of root sampling has limited efforts to understand the effects of N on roots to shoots. Such research has been conducted under pot conditions (*Wang et al., 2019b*); thus, it is not possible to assess variation due to differences in light and temperature conditions as well as nitrate N leaching, as occurs in
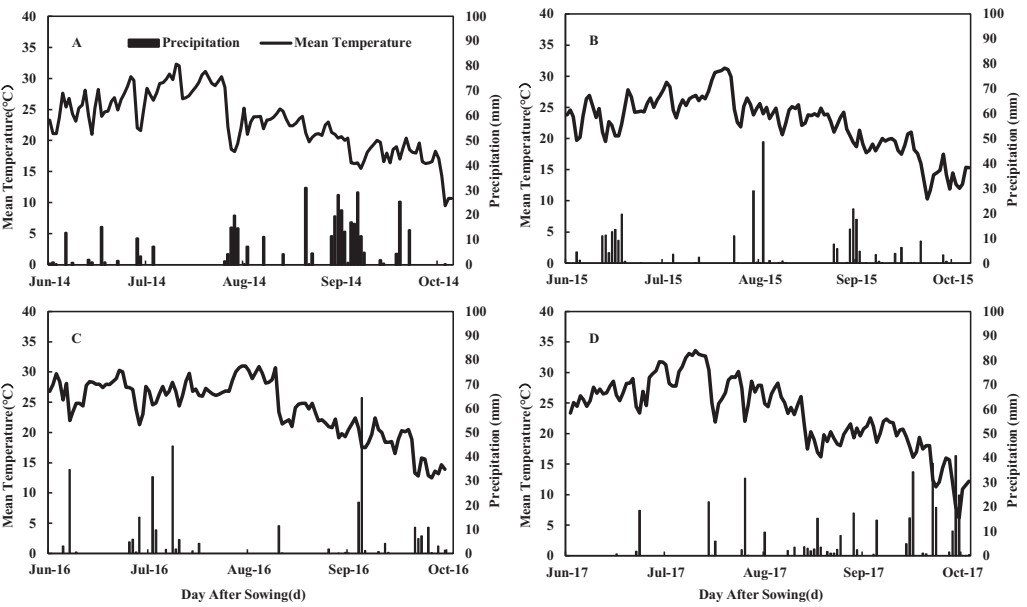

**Figure 1 Daily mean temperature and precipitation during the maize growing seasons in 2014 (A), 2015 (B), 2016 (C) and 2017 (D) at the experimental site.**

field conditions. Limited knowledge about shoot and root traits related to maize grain yield, NUE, RUE, and WUE under field conditions was investigated. Therefore, exploring the response of photosynthetic parameters and root development to grain yield reduction due to N reduction will provide an important reference for management of N fertilizer inputs.

In the current study, we determined that N reduction results in maintenance of photosynthetic activity and root development to maximize grain yield and radiation, water, and N use by maize crops. The objectives of this study were to (1) investigate the effects of N reduction on water, radiation, and N use efficiencies in maize crops, and (2) determine the effects of N reduction on photosynthetic activity and root development in maize crops under semi-arid conditions. Our findings provide key data to support enhanced maize production and resource use efficiency.

# MATERIALS AND METHODS

## Field experiments

A 4-year field experiment was conducted on the same land at the agricultural experimental station of Northwest A&F University in Yangling (34°20′N, 108°04′E, elevation, 455 m), Shaanxi province, China in 2014, 2015, 2016, and 2017. The experimental site experienced an annual average daylight of 2,150 h, an annual average temperature of 12–14 °C, and an average annual precipitation of 581 mm. The annual mean temperature and total precipitation of the experimental area are shown in Fig. 1. The soil of the experimental site is classified as a dark loess soil, and the former crop was winter wheat.

**Table 1 Soil chemical properties of the top 0–60 cm layers in the experimental fields.**

| Layer (cm) | Organic matter (g Kg$^{-1}$) | Total nitrogen (g Kg$^{-1}$) | Alkaline nitrogen (mg Kg$^{-1}$) | Available phosphorus (mg Kg$^{-1}$) | Available potassium (mg Kg$^{-1}$) |
|---|---|---|---|---|---|
| 0–20 | 15.43 | 1.203 | 67.38 | 12.19 | 161.17 |
| 20–40 | 14.02 | 0.987 | 44.89 | 9.03 | 117.21 |
| 40–60 | 13.46 | 1.069 | 45.98 | 6.96 | 102.67 |

Before sowing, soil chemical properties were analyzed in the top 60 cm of soil for organic matter content, N, phosphorus (P), and potassium (K) (Table 1).

The research work was carried out with a randomized complete block design with three replications. The total sub plot size was 39 m$^2$ (7.8 m long and 5 m wide). The row-to-row spacing was maintained at 60 cm and plant-to-plant spacing at 25 cm. The maize seeds were planted manually, and in each hill, two seeds were sown at a depth of 5 cm in the middle of June during each growing season. Plants were thinned manually for normal plant population densities in the area of (67,500 pl. ha) at V3 (three-leaf stage) (*Ritchie & Hanway, 1982*). Plots were kept free of weeds, insects, and diseases; and the weeding was controlled by hand and hoe during each growing season. The N treatments applied in this study were: (1) no N ($N_0$, 100% reduction from $N_{300}$); (2) 150 Kg N ha$^{-1}$ ($N_{150}$, 50% reduction from $N_{300}$); (3) 225 Kg N ha$^{-1}$ ($N_{225}$, 25% reduction from $N_{300}$); and (4) 300 Kg N ha$^{-1}$ ($N_{300,}$ the traditional N dose applied by farmers in the Loess Plateau of China). Fertilizer N was sourced from urea (46% N), evenly split in the fractions of 1/2 at pre-sowing and side-banded deep (5 cm) into the soil on the sowing rows of 1/2 at the twelve-leaf stage. A total of 150 Kg of phosphorus (calcium superphosphate, $P_2O_5$ 16%) ha$^{-1}$ and 150 Kg of potassium (as potassium sulfate, $K_2O$ 45%) ha$^{-1}$ were applied 5 days before sowing. Irrigation was applied at the twelve-leaf stage (75 mm). The amount of irrigation was controlled by a water meter (Zhejiang Ningbo Water Meter Co., Ltd., Ningbo, China).

## Sampling and measurements

### Intercepted photosynthetically active radiation

During the V6–R3, IPAR was measured every 5–7 days (on clear and sunny days between 11:00 AM and 2:00 PM) using a SunScan Canopy Analyzer (Delta, Cambridge, UK). In each plot, three points were chosen. When measuring, the line sensor was placed horizontally between the two ridges (five cm from the soil surface) and was used to collect three consecutive readings of transmitted photosynthetically active radiation (*Chen et al., 2016b*). Round-trip observations were used to minimize error.

### Soil water content

At sowing and maturity, soil water content from the 0 to 200 cm layer was determined using a hand-held soil iron drill (*Zhang et al., 2019*). Between 0 and 20 cm, samples were collected every 10 cm, and between 20 and 200 cm, samples were collected every 20 cm. Soil samples were stored in a closed aluminum box and weighed before drying, oven-dried
at 105 °C for 24 h, and weighed separately. The soil water content of each plot was calculated from the average of three random soil core samples. The soil water content was calculated as the difference between the fresh soil weight minus the dry soil weight divided by the dry soil weight.

### Photosynthetic parameters and leaf area index

In 2016 and 2017, 20 plants were marked prior to the eight-leaf stage (V8). At V8, the tasseling stage (VT), milking stage (R3), and physiological maturity (R6), the three marked plants per plot were selected to measure the net photosynthetic rate (Pn), intercellular $CO_2$ concentration (Ci), and stomatal conductance (Gs) on the ear leaves (at VT, R3, and R6) or fully expanded leaves at the top of a plant (at V8) using a photosynthesis analyzer system (LI-6400, LI-COR, Lincoln, NE, USA) on a clear sunny day between 9:00 AM and 11:00 AM (*Zhang et al., 2017*). The same plants were used for measuring the green leaf area at V8, VT, R3, and R6. The green leaf area index (LAI) was calculated as follows (*Birch, Vos & Van Der Putten, 2003*):

LAI = 0.75 × Leaf length × maximum width × number of plants within a unit area of land/area of land.

## Root system

Three plants root were sampled using the soil profile method (*Holanda et al., 1998*) at the V8, VT, R3, and R6 stages of maize. Each root system was excavated from an area of 0.15 m² (line spacing 0.6 m × row spacing 0.25 m) soil around the center of the plant. Root sampling was conducted at depth intervals of 0–30 cm (surface soil layer), 30–60 cm (middle soil layer), and 60–90 cm (deep soil layer) in each plot. Excavated roots were immersed overnight in a plastic container filled with water and washed with tap water on a 0.25-mm screen until the roots were free of soil. Roots were scanned using an HP Scanjet 8200 scanner, and each root image was analyzed using a root analysis program (WinRhizo Provision 5.0; Regent Instruments Inc., Sainte-Foy, QC, Canada) to obtain the root surface area (cm² plant$^{-1}$) and root length (mm). Root volume was measured by the drainage method. Root length density was calculated as the average of three plants' root lengths divided by the soil volume (*Li et al., 2010*). Root samples were dried for 48h at 70 °C in an oven to obtain the root dry weight per plant.

## Biomass yield, shoot n content, and grain yield

Four central rows were harvested randomly at 20 m² to measure the grain yield at harvest in each plot. Ten ears were randomly selected from each plot and threshed separately to determine moisture content and kernel number. Grain yield was estimated based on kernel weight and water content and expressed as 14% (w/w) moisture content. Six plants were sampled at each plot and were divided into leaves, stems, and grains. Before determining the N concentration, all plant parts were dried at 70 °C for 48 h and weighed. After weighing, the samples were ground into powder using a Wiley-type mill (<1 mm mesh), weighed (0.3–0.4 g), and were mineralized using $H_2SO_4$–$H_2O_2$; then, total N concentration was obtained by using an automatic Kjeldhal microdistillation analyzer (FOSS, Västra Götaland, Sweden, *Nelson & Sommers, 1973*).

## Statistical analysis

Daily intercepted solar radiation was calculated using the following equation (*Liu et al., 2014*):

$$LT = \frac{PAR_L}{PAR_T} \tag{1}$$

where LT is the light transmission ratio, $PAR_L$ is the IPAR at the bottom of the canopy, and $PAR_T$ is the IPAR at the top of the canopy.

The measured IPAR value of the bottom layer was analyzed by two-dimensional interpolation over 1 day to obtain the IPAR of the entire canopy. Then, the IPAR rate obtained by interpolation analysis is multiplied by the incident PAR measured on the corresponding date by the meteorological observatory to determine the amount of canopy IPAR.

Radiation use efficiency was calculated using the following equation:

$$RUE = \frac{Mh}{\sum Q} \times 10^{-7} \times 100\% \tag{2}$$

where $\sum Q$ (MJ m$^{-2}$) is the accumulated intercepted solar radiation, h (KJ Kg$^{-1}$) is the heat energy released of per Kg of grain yield, and M is the grain yield (Kg ha$^{-1}$).

Water use efficiency was calculated as:

$$WUE = \frac{GY}{ET} \tag{3}$$

where GY (Kg ha$^{-1}$) is the grain yield, and ET (mm) is the evapotranspiration, as calculated as by soil water balance equation (*Huang et al., 2005*).

The following equations were also used:

Internal N efficiency,

$$INE = \frac{GY}{SNC} \tag{4}$$

Agronomic N use efficiency,

$$ANE = \frac{(GY_{Ni} - GY_{N0})}{Ni} \tag{5}$$

Apparent N recovery efficiency,

$$REN \ (\%) = \frac{(SNC_{Ni} - SNC_{N0})}{Ni} \times 100 \tag{6}$$

N harvest index,

$$NHI \ (\%) = \frac{GNC}{SNC} \times 100 \tag{7}$$

where SNC (Kg ha$^{-1}$) is the shoot N content calculated as biomass (Kg ha$^{-1}$) multiplied by shoot N concentration (Kg Kg$^{-1}$), i (N rates, Kg ha$^{-1}$) is N application rates 150, 225, or 300, $GY_{Ni}$ (Kg ha$^{-1}$) is grain yield in the N application plots, $GY_{N0}$ (Kg ha$^{-1}$) is grain yield

in the no-N application plots, and Ni is the N application rate. $SNC_{Ni}$ (Kg ha$^{-1}$) is the shoot N content in N application plots, $SNC_{N0}$ (Kg ha$^{-1}$) is the shoot N content in the no-N application plots. GNC is grain N content (Kg ha$^{-1}$).

The experimental data were organized and processed using Microsoft Excel 2013 and are presented with standard error. SPSS18.0 (SPSS Institute Inc., Chicago, IL, USA) statistical analysis software was used for variance analysis. The data was checked for normality (Kolmogorov–Smirnov test) and homogeneity of variance (Bartlett–Box test). The effects of N rates, years, and their interactions on the measured variables were tested using one- and two-way ANOVAs. To identify significant treatments effects, multiple comparisons among different treatments were performed using Duncan's multiple range test. Differences with $p < 0.05$ were considered statistically significant.

# RESULTS

## Grain yield, biomass yield, and crop resource utilization

Our results revealed that the year and N application rates showed significant effects on grain yield (GY), biomass yield (BY), grain N content (GNC), agronomic N use efficiency (ANE), apparent N recovery efficiency (REN), and N harvest index (NHI) (Table 2). The interaction between N application rate and year had no significant effect on the above parameters. We observed no significant differences in GY, BY, and GNC between $N_{225}$ and $N_{300}$ in all growing seasons. Compared with $N_{300}$, the GY of $N_{150}$ and $N_{0}$ decreased ranging from 4.7% to 13.6% and 19.7% to 22.8%, respectively, while the grain yield of treatment $N_{225}$ increased ranging from 1.5% to 3.7% averaged of 4 years. Treatment $N_{300}$ increased BY ranging from 28% to 34%, compared with $N_{0}$, while $N_{225}$ increased BY ranging from 27% to 32% in the 4 years. The GY difference among years may be due to the rainfall amount and seasonal distribution. Rainfall was 390, 284, 311, and 371 mm in 2014, 2015, 2016, and 2017, respectively (Fig. 1). Among the experimental years, GY was higher in 2016 and 2017 compared to 2014 and 2015. 2017 was a more suitable year for maize growth. Although the rainfall in 2016 was reduced, the distribution was relatively uniform throughout the growth period. The early rainfall ensured the regularity and vegetative growth of maize seedlings. The lower GY in 2015 can be explained by the reduced rainfall. In 2014, the greater rainfall was mainly due to the large amount of rainfall occurring 70 days after sowing. Continuous rainfall from the silking to the flowering stage affected maize pollination, and severe stalk rot disease occurred during the grain filling stage, which caused pro-senescence (Fig. 1), causing lower GY. The optimal rate of N application increased the AEN, REN, and NHI, except for the AEN of $N_{150}$ during the 2016 growing season. The NHI was highest for the $N_{225}$ and $N_{150}$ (2014, 2015, and 2017) or $N_{225}$ (2016) treatments.

Throughout the crop growth cycle, differences in ET among N treatments were significant ($p < 0.05$, Table 3). During growing seasons 2015, 2016, and 2017, we observed that the rate of N application showed no significant effect on IPAR. The IPAR of $N_{0}$ was less than that of $N_{300}$ in the 2014 growing season. SNC was increased by increasing the N application rate. For $N_{300}$, SNC increased ranging from 73% to 92% compared with $N_{0}$, and $N_{225}$ application increased SNC ranging from 68% to 81% compared with the $N_{0}$

**Table 2** Effects of nitrogen application rates on grain yield (GY), biomass yield (BY), grain nitrogen content (GNC), N harvest index (NHI) and agronomic N use efficiency (AEN), apparent N recovery efficiency (RNE).

| Year | Nitrogen rates | GY (t ha$^{-1}$) | BY (Kg ha$^{-1}$) | GNC (Kg ha$^{-1}$) | NHI | AEN (Kg Kg$^{-1}$) | REN (Kg Kg$^{-1}$) |
|---|---|---|---|---|---|---|---|
| 2014 | $N_{300}$ | 10.2a | 18.1a | 127a | 59.4b | 7.6b | 38.4b |
| | $N_{225}$ | 10.5a | 17.9a | 128a | 64.5a | 11.5a | 44.5a |
| | $N_{150}$ | 9.4a | 16.4b | 105b | 64.3a | 9.9ab | 43.1ab |
| | $N_0$ | 7.9b | 13.6c | 61c | 62.0ab | – | – |
| 2015 | $N_{300}$ | 10.7a | 18.6a | 134a | 63.1b | 8.1b | 35.3b |
| | $N_{225}$ | 11.1a | 18.6a | 133a | 67.3a | 12.5a | 41.0a |
| | $N_{150}$ | 10.2a | 17.3b | 111b | 66.1a | 12.8a | 41.3a |
| | $N_0$ | 8.3b | 14.4c | 68c | 64.7b | – | – |
| 2016 | $N_{300}$ | 11.3a | 20.0a | 142a | 60.5b | 8.5b | 38.7b |
| | $N_{225}$ | 11.4a | 19.8a | 134a | 63.6a | 12.1a | 41.6ab |
| | $N_{150}$ | 9.7b | 18.4b | 113b | 59.0b | 6.9b | 48.6a |
| | $N_0$ | 8.7c | 15.6c | 70c | 59.1b | – | – |
| 2017 | $N_{300}$ | 11.0a | 20.0a | 137a | 61.4b | 7.2b | 37.2b |
| | $N_{225}$ | 11.4a | 20.0a | 138a | 65.1a | 11.5a | 45.0a |
| | $N_{150}$ | 10.5a | 18.5b | 116b | 63.2ab | 11.1a | 47.7a |
| | $N_0$ | 8.8b | 15.6c | 70c | 62.7b | – | – |
| Source of variation | | | | | | | |
| | Nitrogen rates (N) | ** | ** | ** | ** | ** | ** |
| | Year (Y) | ** | ** | ** | ** | ** | ** |
| | N × Y | ns | ns | ns | ns | ns | ns |

**Note:**
Means followed by different lowercase letters indicate significantly different ($p < 0.05$) within the same column and the same year. $N_{300}$, $N_{225}$, $N_{150}$ and $N_0$ represent application of nitrogen at a rate of 300, 225, 150 and 0 Kg ha$^{-1}$. **Significant at the 0.01 probability level. ns, non-significant

treatment in the 4 years. IEN was increased with a low rate N application. Application of N at 225 Kg ha$^{-1}$ increased IEN ranging from 8.5% to 12.3% compared with $N_{300}$, while $N_{150}$ decreased IEN ranging from 9.7% to 20.7% compared with $N_{300}$ in the 4 years. N application at the $N_{300}$ and $N_{225}$ rates were similar, while $N_{300}$ increased RUE ranging from 19% to 26%, whereas $N_{225}$ increased RUE ranging from 23% to 29% compared with $N_0$ in the 4 years. Reduced N application was associated with reduced WUE. Our results showed that $N_{225}$ increased grain WUE ranging from 25% to 26% compared to $N_0$ in the 4 years. WUE was also significantly increased in the $N_{300}$ treatment but to a lesser extent (19–22%). RUE and NUE exhibited non-linear responses to the N application rate, indicating that the maximum grain yield can occur with $N_{150}$ or $N_{225}$ treatments. The differential grain yield between $N_{225}$ and $N_{300}$ was relatively small.

## Photosynthetic parameters and LAI

High external nitrogen application rates and year had significant effects on Pn, Gs, and Ci, but the interaction between N application rates and year did not display significant

**Table 3 Effect of nitrogen application rates on evapotranspiration (ET), the accumulated intercepted solar radiation (IPAR), shoot nitrogen content (SNC), internal N use efficiency (INE), radiation use efficiencies (RUE), and water use efficiency (WUE).**

| Year | Nitrogen rate | ET (mm) | IPAR (MJ m$^{-2}$) | SNC (Kg ha$^{-1}$) | IEN (Kg Kg$^{-1}$) | RUE | WUE |
|---|---|---|---|---|---|---|---|
| 2014 | $N_{300}$ | 350a | 974a | 214a | 47.8c | 1.96a | 29.1a |
| | $N_{225}$ | 346a | 960a | 198a | 53.1b | 2.05a | 30.4a |
| | $N_{150}$ | 343a | 989a | 163b | 57.6b | 1.77b | 27.4b |
| | $N_0$ | 333b | 906b | 98c | 81.6a | 1.63c | 23.8c |
| 2015 | $N_{300}$ | 336a | 1181a | 212a | 50.5d | 1.69a | 31.8a |
| | $N_{225}$ | 337a | 1179a | 198a | 55.9c | 1.75a | 32.9a |
| | $N_{150}$ | 317ab | 1131a | 168b | 60.8b | 1.68a | 32.1a |
| | $N_0$ | 303b | 1138a | 106c | 78.1a | 1.35b | 27.2b |
| 2016 | $N_{300}$ | 414a | 1141a | 234a | 48.1c | 1.84a | 27.2a |
| | $N_{225}$ | 416a | 1124a | 212a | 54.0b | 1.89a | 27.5a |
| | $N_{150}$ | 399ab | 1120a | 191b | 51.0b | 1.62b | 24.4b |
| | $N_0$ | 396b | 1102a | 118c | 74.2a | 1.47c | 22.0c |
| 2017 | $N_{300}$ | 420a | 1134a | 223a | 49.5d | 1.82a | 26.3ab |
| | $N_{225}$ | 417a | 1121a | 213a | 53.7c | 1.91a | 27.5a |
| | $N_{150}$ | 409ab | 1122a | 183b | 57.5b | 1.75b | 25.8b |
| | $N_0$ | 392b | 1100a | 112c | 79.4a | 1.50c | 22.6c |
| Source of variation | | | | | | | |
| | Nitrogen (N) | * | * | ** | ** | ** | ** |
| | Year (Y) | ** | ** | ** | * | * | ** |
| | N × Y | ns | ns | ns | ns | ns | ns |

Note:
Means followed by different lowercase letters within each column indicate significantly different ($p < 0.05$). $N_{300}$, $N_{225}$, $N_{150}$ and $N_0$ represent application of nitrogen at a rate of 300, 225, 150 and 0 Kg ha$^{-1}$. ET, total evapotranspiration (mm); IPAR, the accumulated intercepted solar radiation (MJ m$^{-2}$), SNC, shoot nitrogen content (Kg ha$^{-1}$), IEN, internal N use efficiency (Kg Kg$^{-1}$); RUE, radiation use efficiency; WUE, water use efficiency. *Significant at the 0.05 probability level. **Significant at the 0.01 probability level. ns, non-significant.

correlated (Fig. 2). In both 2016 and 2017, the $N_0$ and $N_{150}$ treatments resulted in lower values of Pn and Gs when compared with $N_{225}$ and $N_{300}$; however, the effect of $N_{225}$ was relatively greater than that of the $N_{300}$ treatment at R6. This finding indicates that N fertilizer inputs can increase Gs and improve the photosynthetic capacity of maize crops. Conversely, the Pn and Gs of the $N_{300}$ treatment were reduced compared with $N_{225}$. Pn of $N_{150}$, $N_{225}$, and $N_{300}$ increased ranging from 16% to 81% across growth stages compared with $N_0$ averaged of 2 years. N application rates significantly affected 1-Ci/Ca ($p < 0.05$). The 1-Ci/Ca of the $N_0$ treatment was significantly lower than that of $N_{300}$ in all measurements (Fig. 2).

Overall, the LAI was unaffected ($p = 0.55$) by N application rates, with an average of 2.6 and 3.7 at the V8 and VT stages (Fig. 2). N application rates significantly affected LAI at the R3 and R6 stages. Compared with $N_{300}$, the LAI of $N_{225}$, $N_{150}$ and $N_0$ decreased by 1%, 6%, 11%, respectively, and 1%, 7%, and 13%, respectively, at the R3 and R6 stages.

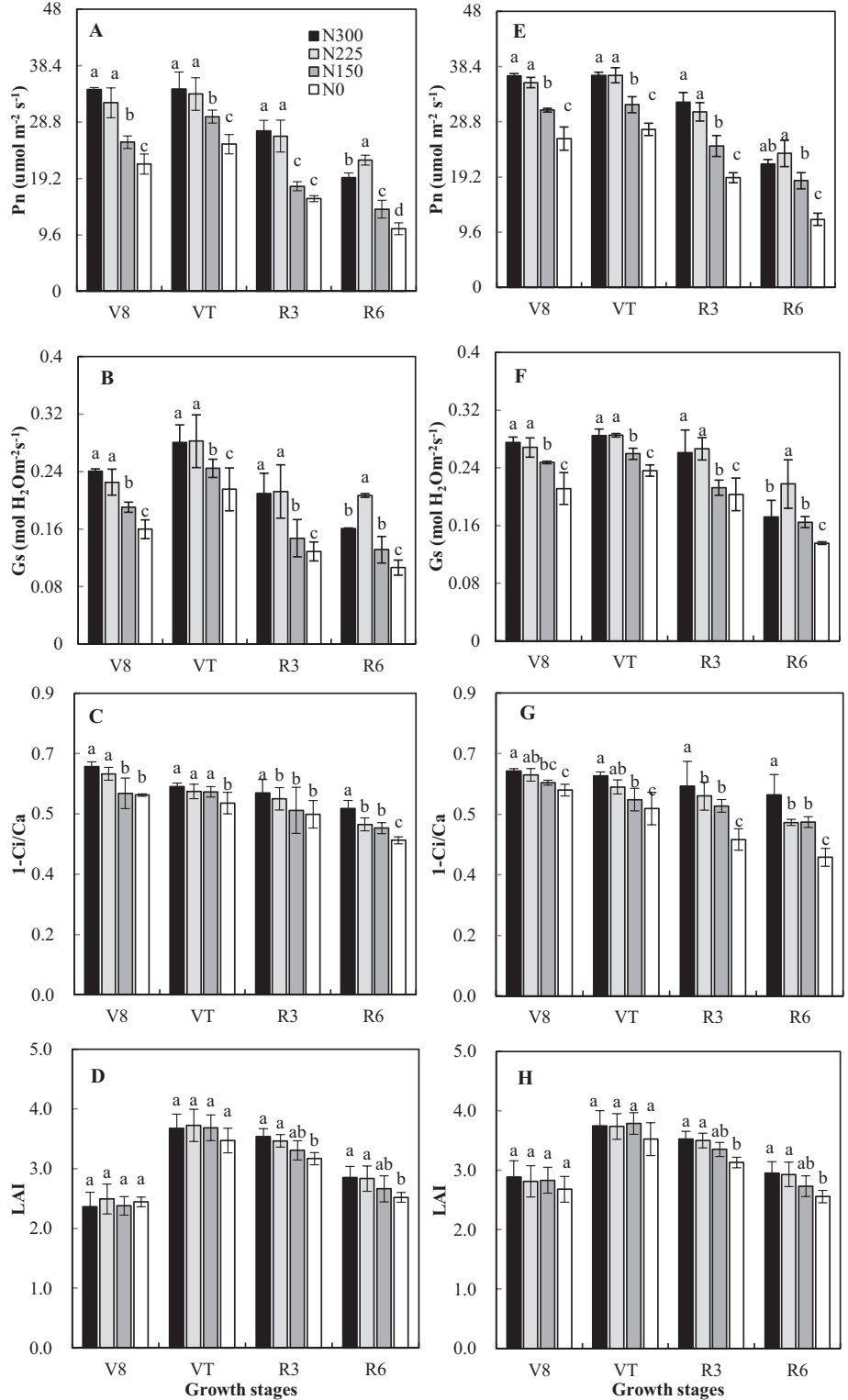

**Figure 2 Effects of nitrogen application rates on Pn, Gs, 1-Ci/Ca and leaf area index (LAI) in 2016 (A–D) and 2017 (E–H) growing seasons.** $N_{300}$, $N_{225}$, $N_{150}$ and $N_0$ represent application of nitrogen at a rate of 300, 225, 150 and 0 Kg ha$^{-1}$. Vertical bars represent the (means ± SD) ($n = 3$).

**Table 4 Effect of nitrogen application rates on root dry matter and root ratio.**

| Year | Soil layer | Nitrogen rate | Root dry matter (g plant$^{-1}$) | | | | Root ratio % | | | |
|---|---|---|---|---|---|---|---|---|---|---|
| | | | V8 | VT | R3 | R6 | V8 | VT | R3 | R6 |
| 2016 | 0–30 | $N_{300}$ | 2.81a | 19.89b | 19.09ab | 16.45b | 0.99 | 0.93 | 0.93 | 0.93 |
| | | $N_{225}$ | 3.05a | 20.78a | 20.43a | 17.34a | 0.99 | 0.92 | 0.92 | 0.91 |
| | | $N_{150}$ | 1.53b | 20.18b | 17.50b | 14.41b | 0.99 | 0.94 | 0.92 | 0.9 |
| | | $N_0$ | 0.97c | 15.59c | 13.19a | 10.37c | 1 | 0.89 | 0.91 | 0.9 |
| | 30–60 | $N_{300}$ | 0.03a | 1.52b | 1.22b | 1.23bc | 0.01 | 0.07 | 0.06 | 0.07 |
| | | $N_{225}$ | 0.03a | 1.65a | 1.45a | 1.45a | 0.01 | 0.07 | 0.07 | 0.08 |
| | | $N_{150}$ | 0.01b | 1.30c | 1.26b | 1.26b | 0.01 | 0.06 | 0.07 | 0.08 |
| | | $N_0$ | 0.00b | 1.49b | 1.22b | 0.99c | 0 | 0.08 | 0.08 | 0.09 |
| | 60–90 | $N_{300}$ | | 0.03b | 0.13b | 0.08c | 0 | 0 | 0.01 | 0 |
| | | $N_{225}$ | | 0.05b | 0.32a | 0.34a | 0 | 0 | 0.01 | 0.02 |
| | | $N_{150}$ | | 0.07b | 0.26a | 0.26b | 0 | 0 | 0.01 | 0.02 |
| | | $N_0$ | | 0.52a | 0.12b | 0.12c | 0 | 0.03 | 0.01 | 0.01 |
| 2017 | 0–30 | $N_{300}$ | 3.06b | 22.09a | 19.90ab | 17.33b | 0.96 | 0.93 | 0.93 | 0.92 |
| | | $N_{225}$ | 4.67a | 22.99a | 21.90a | 18.80a | 0.97 | 0.93 | 0.92 | 0.91 |
| | | $N_{150}$ | 2.03c | 21.99a | 19.42b | 15.81b | 0.98 | 0.94 | 0.91 | 0.9 |
| | | $N_0$ | 1.36c | 15.32b | 14.20c | 11.11c | 1 | 0.88 | 0.9 | 0.89 |
| | 30–60 | $N_{300}$ | 0.12a | 1.52b | 1.32c | 1.24c | 0.04 | 0.06 | 0.06 | 0.07 |
| | | $N_{225}$ | 0.14a | 1.65a | 1.61a | 1.53a | 0.03 | 0.07 | 0.07 | 0.07 |
| | | $N_{150}$ | 0.03b | 1.30c | 1.48b | 1.42b | 0.02 | 0.06 | 0.07 | 0.08 |
| | | $N_0$ | 0.01b | 1.49b | 1.36c | 1.19c | 0 | 0.09 | 0.09 | 0.1 |
| | 60–90 | $N_{300}$ | | 0.03b | 0.21c | 0.19b | 0 | 0 | 0.01 | 0.01 |
| | | $N_{225}$ | | 0.05b | 0.40a | 0.41a | 0 | 0 | 0.02 | 0.02 |
| | | $N_{150}$ | | 0.07b | 0.34b | 0.33b | 0 | 0 | 0.02 | 0.02 |
| | 60–90 | $N_0$ | | 0.65a | 0.21c | 0.12b | 0 | 0.04 | 0.01 | 0.01 |

Note:
Means followed by different lowercase letters within each column indicate significantly different ($p < 0.05$). $N_{300}$, $N_{225}$, $N_{150}$ and $N_0$ represent application of nitrogen at a rate of 300, 225, 150 and 0 Kg ha$^{-1}$.

## Root system

The root dry weight reached a maximum value with the $N_{225}$ treatment. Root dry weight of $N_{225}$ increased ranging from 35% to 67% compared with $N_0$, while $N_{300}$ increased the root dry weight ranging from 29% to 53% compared with $N_0$ average of 2 years (Table 4). We observed clear differences in the vertical distributions of roots as well as root morphology between N application rates, including root volume, root surface area, and root length density, which varied with N application rates and years (Table 5). After the VT stage, the root system distributed in the surface layer (0–30 cm) accounted for greater than 88% of the root weight for all treatments. At the V8 stage, the root ratio was proportional to the N application rate and increased significantly in the middle layer (30–60 cm). In the $N_0$ treatment, almost all roots were concentrated in the surface soil layer. The $N_0$ treatment exhibited significant increases in the deep-layer (60–90 cm) root ratio at the VT stage. The root ratio in the deep soil layer decreased with increasing

**Table 5 Effect of nitrogen application rates on root volume, root surface and root length density.**

| Year | Soil layer | Nitrogen rate | Root volume (cm³ plant⁻¹) | | | | Root surface (cm² plant⁻¹) | | | | Root length density (mm cm⁻³) | | | |
|---|---|---|---|---|---|---|---|---|---|---|---|---|---|---|
| | | | V8 | VT | R3 | R6 | V8 | VT | R3 | R6 | V8 | VT | R3 | R6 |
| 2016 | 0–30 | N300 | 31.2a | 80.7ab | 80.7a | 68.6b | 306.19a | 536.5b | 695.8b | 569.3b | 0.86a | 1.24b | 0.82b | 1.52b |
| | | N225 | 37.7a | 91.2a | 89.9a | 76.6a | 339.78a | 611.6a | 744.9a | 639.5a | 0.98a | 1.54a | 2.30a | 1.78a |
| | | N150 | 17.3b | 86.8b | 64.0b | 52.4c | 167.56b | 652.4a | 563.5c | 472.6c | 0.50b | 1.23b | 1.54c | 1.11c |
| | | N0 | 10.7c | 54.0c | 31.7c | 26.9d | 132.53c | 405.1a | 281.8d | 228.5d | 0.40b | 0.90c | 0.70d | 0.53d |
| | 30–60 | N300 | 1.9a | 6.7b | 6.2a | 2.9b | 10.53a | 101.4b | 73.9b | 51.1b | 0.05a | 0.34bc | 0.48a | 0.18b |
| | | N225 | 0.8b | 9.0a | 6.7a | 5.5a | 8.94a | 117.4a | 138.0a | 100.2a | 0.03a | 0.39b | 0.54a | 0.4a |
| | | N150 | 0.1c | 4.2c | 3.3b | 2.6b | 0.97b | 88.5b | 90.8b | 66.5b | 0.003b | 0.31c | 0.44b | 0.35a |
| | | N0 | 0.06c | 7.1b | 2.9b | 1.9c | 0.17c | 112.5a | 14.7c | 36.5c | 0.000b | 0.46a | 0.22c | 0.13b |
| | 60–90 | N300 | | 1.1b | 2.2b | 0.9c | | 11.1c | 44.1c | 26.2c | | 0.01c | 0.19c | 0.07b |
| | | N225 | | 3.2a | 5.2a | 3.9a | | 37.4b | 118.0a | 87.9a | | 0.17b | 0.39a | 0.24a |
| | | N150 | | 1.5b | 3.1b | 2.6b | | 26.1b | 91.0b | 72.1b | | 0.07c | 0.25b | 0.21a |
| | | N0 | | 4.6a | 2.0b | 1.0c | | 118.0a | 47.9c | 31.3c | | 0.35a | 0.12c | 0.09b |
| 2017 | 0–30 | N300 | 36.8a | 92.3b | 85.6b | 71.6b | 428.83b | 624.4b | 726.8a | 611.8b | 1.08b | 1.37b | 1.77b | 1.29b |
| | | N225 | 42.9a | 112.9a | 103.8a | 94.3a | 520.53a | 683.4a | 779.8a | 690.1a | 1.12a | 1.74a | 3.05a | 2.67a |
| | | N150 | 21.2b | 100.4a | 73.9b | 63.5b | 248.31c | 690.1a | 609.6b | 487.3c | 0.56c | 1.43b | 1.92b | 1.54b |
| | | N0 | 11.8c | 58.4c | 38.4c | 32.6c | 166.37d | 560.2c | 345.4c | 286.7d | 0.52d | 1.00c | 0.97c | 0.58c |
| | 30–60 | N300 | 2.7a | 8.3b | 7.2b | 3.3b | 11.37a | 115.7a | 100.0b | 64.2c | 0.05a | 0.49b | 0.57a | 0.33b |
| | | N225 | 1.2b | 11.0a | 8.4a | 6.6a | 14.15a | 133.0a | 168.5a | 146.1a | 0.06a | 0.53a | 0.60a | 0.52a |
| | | N150 | 0.4c | 5.5c | 4.4bc | 3.4b | 1.91b | 110.4a | 110.2b | 96.0b | 0.02b | 0.44b | 0.53b | 0.48a |
| | | N0 | 0.3c | 8.5b | 3.8c | 2.6b | 0.34c | 145.5b | 29.0c | 54.9c | 0.03b | 0.58a | 0.33c | 0.25c |
| | 60–90 | N300 | | 1.4b | 3.2b | 1.1c | | 29.3d | 76.2b | 49.1c | | 0.09c | 0.29c | 0.10bc |
| | | N225 | | 4.8a | 6.3a | 4.3a | | 84.3b | 145.7a | 126.3a | | 0.30b | 0.47a | 0.31a |
| | | N150 | | 2.3b | 3.7b | 3.1b | | 46.2c | 101.1b | 87.9b | | 0.13c | 0.36b | 0.30b |
| | | N0 | | 5.5a | 3.0b | 2.0c | | 145.1a | 60.2c | 36.3c | | 0.44a | 0.20c | 0.10c |

Note:
Means followed by different lowercase letters within each column indicate significantly different ($p < 0.05$). $N_{300}$, $N_{225}$, $N_{150}$ and $N_0$ represent application of nitrogen at a rate of 300, 225, 150 and 0 Kg ha⁻¹.

N application after the VT stage. At the R3 and R6 stages, the surface root ratio increased with increasing N application rate. For the $N_0$ treatment, the deep-layer root ratio decreased after the VT stage. Similar to the dry root weight, root morphology indexes (root volume, surface area, and length density) increased initially and then decreased with increasing N application rate. These indexes reached their maximum values under the $N_{225}$ treatment. At the V8 stage, root morphology indexes were in the order of $N_{225} > N_{300} > N_{150} > N_0$ in the surface soil layer, which differed from the middle soil layer ($N_{300} > N_{225} > N_{150} > N_0$). Larger roots appeared in deep soil layers at the VT stage when compared to the V8 stage, and the maximum value (root volume, surface area, and length density) for the $N_0$ treatment was observed in the middle or deep soil layers. At the R3 and R6 stages, trends in root morphology indexes varied between soil layers; in the surface layer, the indexes followed the order $N_{225} > N_{300} > N_{150} > N_0$; in the middle layer, $N_{225} > N_{150} > N_{300} > N_0$; and in the deep layer, $N_{225} > N_{150} > N_0 > N_{300}$.

## DISCUSSION

An optimal rate of N application is expected to produce a balance between crop demand and N supply and ensure maximum crop production while conserving resources and protecting against environmental damage (*Ciampitti & Vyn, 2011*; *Peng, Li & Fritschi, 2014*). Our results showed that during the 2017 growing season, even when photosynthesis was significantly affected by a 50% reduction in N application ($N_{150}$), grain yield was not significantly decreased. In 2016, photosynthetic parameters decreased further for the $N_{150}$ and $N_0$ treatments, and the grain yield also decreased significantly. Over 4 years, we observed no significant reduction in grain yield for the $N_{225}$ treatment (Table 2), which even displayed a grain yield higher than that of the $N_{300}$ treatment, although the difference was not significant. Our research indicated that the N dose could be reduced by at least 25% without compromising grain production. Our results are similar to those of *Li et al. (2007)* and *Lamptey et al. (2017),* who reported an optimal N fertilization range for summer maize of 200 to 300 Kg N ha$^{-1}$. Under the $N_0$ and $N_{150}$ treatments, dry matter accumulation was limited, and the difference between $N_{300}$ and $N_{225}$ treatments was not significant. The reduction in crop yield induced by N reduction can be explained by various factors, as described below.

Our experiments also addressed the mechanism by which maize yield is affected by N application rates. We found that for most years, IPAR was not affected by N fertilizer application rates, a finding similar to that described in previous studies (*Vos, Van Der Putten & Birch, 2005*; *Massignam et al., 2012*). N application rate significantly affected RUE ($p < 0.05$). For $N_0$ and $N_{150}$, the RUE was significantly less than that of $N_{300}$. However, the RUE of the $N_{225}$-treated crop was greater than that of the $N_{300}$-treated crop. In this case, lower grain yield driven by lower N treatment corresponded to lower RUE. These results indicate that under such production conditions, N application rate mainly affects grain yield by affecting RUE rather than IPAR. *Vos, Van Der Putten & Birch (2005)* also found that maize tends to sacrifice specific leaf N and RUE while maintaining leaf area (small changes to LAI (Fig. 2) in comparison with the large decrease in Pn (Fig. 2) at low N application rates). Therefore, in the condition of high IPAR, improving the RUE may represent a valid mechanism to achieve high maize grain yield.

The actual ET involves two components: crop transpiration and soil evaporation. N application can increase ET during the reproductive period due to high leaf transpiration under high N conditions (*Lamptey et al., 2017*; *Rudnick et al., 2017*). The relatively low sensitivity of IPAR to N supply in maize may also be consistent with the low sensitivity of soil evaporation to N supply. In this case, the lower grain yields associated with the $N_0$ treatments corresponded to lower ET and WUE values. In addition, WUE of the $N_{300}$-treated crop was lower than that of the $N_{225}$-treated crop. This is due to a higher sensitivity to soil water by the plant at higher N application rates. An increase in N application rates is usually accompanied by a decrease in NUE (*Ju et al., 2015*). Significant effects of N were also observed on N use efficiency (NUE including AEN, REN, NHI, and INE) in our study. Conversely, reducing the N application rate can create a balance between crop demand and N supply (*Lamptey et al., 2017*). The desired

N concentration of the plant, under the $N_0$ treatment, has a high INE value, indicating that the plant N concentration or yield are low and that the amount of N in the plant absorbed from the soil is small. N accumulation increased with increasing N application, but N accumulation in the grain was not significantly different between $N_{225}$ and $N_{300}$ treatments, and the value of NHI at $N_{225}$ was greater than that at $N_{300}$, which indicated that increased N in the plant did not transfer to the grain, resulting in excessive N absorption and residual N in the vegetative organs.

The plasticity of root morphology allows it to respond to soil mineral nutrients (*Peng, Li & Li, 2012*; *Yu et al., 2014*). We found that root dry weight reached a maximum with the $N_{225}$ treatment, which suggested that the relationship between N input and the root system is not linear and positive; N input may even have a negative impact on root growth and development. In the present study, application of N fertilizer promoted growth of roots in the 0–60 cm soil layer and increased the proportion of roots in this layer, indicating that N application improved growth of the upper layer roots. In the late growth stage (after VT), the $N_0$ treatment exhibited a negative effect on the root dry weight and proportion in the 30–60 cm and 60–90 cm soil layers, indicating that N deficiency would be detrimental during the accelerated aging of deep layer roots. Not only are root morphology and nutrient absorption closely related, but the spatial distribution of roots is also closely connected to crop growth and productivity (*Mi et al., 2010*; *Lynch, 2013*). In both years, the root system exhibited the optimal distribution under the $N_{225}$ treatment, with a higher root length density in the observed soil layer, resulting in larger and deeper infiltration scales. Slower root senescence in the $N_{225}$ treatment is also a major contributor to N rate–induced increases in grain yield. Studies have shown that the relative stability of the deep root environment is beneficial in promoting the buffer capacity of the root system in adverse soil environments and achieving high grain yield and resource use efficiency (*Chen et al., 2010*; *Mi et al., 2010*; *Wasson et al., 2012*; *Saengwilai et al., 2014*). The results of our study during the 2016 and 2017 growing seasons showed that excessive N ($N_{300}$) application negatively affects early deep root growth compared with $N_{225}$. High external N input also appeared to generate an overall inhibitory effect on later root growth. There are many reasons for the observed reduction in crop yield. Slight reductions in crop yield induced by excessive N application may be due to negative impacts on root growth during the early growth stage or may be caused by differing mechanisms of aging leading to N loss and relative N deficiency during the reproductive stage. N deficiency induces root thinning and increases longitudinal expansion by promoting root growth in the deep soil, while high N inhibits vertical expansion of roots (*Trachsel et al., 2013*; *Mu et al., 2015*). This study explained the effect of excessive N and N deficiency on yield from the perspective of root morphology and growth.

The role of N in grain formation is mainly explained by photosynthesis, and N reduction usually negatively impacts photosynthetic performance in maize (*Massignam et al., 2012*; *Olszewski et al., 2014*). Although the $N_{150}$ treatment significantly reduced Pn in the 2017 growing season, the grain yield for the $N_{150}$ treatment was not significantly

reduced, indicating that the plant's transient photosynthetic parameters were more sensitive than the dry mass factors in responding to changes in N application rates. Our results illustrate the physiological basis for utilizing the 225 Kg ha$^{-1}$ N rate to improve the stress resistance of summer maize plants in the Loess Plateau. For the $N_{300}$ treatment, decreased Pn was due to lower Gs, which may be due to N-associated increased sensitivity of plant to soil water and results in lower WUE than the $N_{225}$ treatment at the R6 stages. In the low-N treatment, maize plants showed reduced leaf rolling under soil water stress compared with the high N treatment and thus obtained higher WUE (*Wang et al., 2019b*).

At different measurement dates, the maximum value of Gs was not always associated with the $N_{300}$ treatment (Fig. 2). In general, plants' water requirements are expected be greater under high N conditions. Thus, high-N fertilizer plots are more likely to be water-deficient if the soil moisture is inadequate, which aggravates the plant's stomatal limitations (1-Ci/Ca) and resulting in reduction of Gs and Pn. Similar findings have been described in wheat (*Zhang et al., 2017*). Previous studies also pointed out that under soil water-stress, ABA (as the signal carrier) transmitted to the shoot, reducing the Gs and increasing stomatal limitations in the initial drought (*Larcher, 2003*; *Li & Xu, 2014*; *Yan et al., 2017*). The inadequate soil moisture of the $N_{300}$ treatment is not serious enough to affect non-stomatal factors restricting photosynthetic carbon assimilation. The Pn of the $N_{225}$ treatment was similar to that of the $N_{300}$ treatment during the VT stage, but it was less than that of the $N_{225}$ treatment in R6. Although the $N_{225}$ treatment showed a higher photosynthetic rate in the R6 than the $N_{300}$ treatment, the ratio did not result in a significantly higher yield, which may be caused by the small contribution of high photosynthetic capacity to a grain yield (*Acciaresi et al., 2014*). In addition, the photosynthetic capacity may also be related to the differences in N nutrition characteristics among different maize genotypes (*Chen et al., 2014*; *Li et al., 2015*). Thus, while the application rate of 225 Kg N ha$^{-1}$ was may still be high, the current N rate was relatively effective in improving resource use efficiencies. Under production conditions, large amounts of N input are often used as an "insurance" against higher yields to ensure further increases in maize production (*Huang et al., 2007*; *Chen et al., 2012*). However, this behavior results in a significant reduction in NUE (*Chen et al., 2010*) and a slight reduction of RUE and WUE. Previously, *Chen et al. (2016a)* and *Mu et al. (2018)* reported that N is used primarily for cell morphogenesis, that N in leaves is N redundant, and excess N is mainly stored in soluble protein and light-harvesting pigment-protein complexes. Therefore, in our study, the grain yield of $N_{225}$ treatment did not display significantly different results compared to the $N_{300}$ treatment, and the root system and photosynthetic capacity showed certain advantages of $N_{225}$ treatment. From these findings, we can conclude that it is achievable to improve resource use efficiencies while ensuring grain yield. Actually, maize genotypes and soil moisture also affect GY and physiological characteristics (*Ciampitti et al., 2013*). In the current experiment, these factors are not taken into consideration. In future studies, we will focus on the effects of genotype with the aim of maximizing GY and resource use efficiency.

## CONCLUSIONS

Decreased grain yield due to N reduction was driven by reduced radiation utilization efficiency and WUE; the impact of radiation interception and total water evapotranspiration were relatively small. An application rate of 225 Kg N ha$^{-1}$ could be used as a reference for optimal N application in the Loess Plateau of China. This N application rate optimized the eco-physiological responses of the plant, a finding which was confirmed by measuring photosynthetic activity and the root system. This response to optimizing N input resulted in higher grain yield, RUE, WUE, and NUE. Reducing N application rates has strong recoverability in maize production and can maximize the capture and utilization of resources, increasing the maize grain yield.

### Funding

This work was supported by the National High-Tech Research and Development Programs of China ("863 Program") for the 12th Five-Year Plants (No. 2013AA102902), Agro-scientific Research in the Public Interest under Grant (201303104), and the National Natural Science Foundation of China (No. 31601256). The funders had no role in study design, data collection and analysis, decision to publish, or preparation of the manuscript.

### Grant Disclosures

The following grant information was disclosed by the authors:
National High-Tech Research and Development Programs of China: 2013AA102902.
Agro-scientific Research: 201303104.
National Natural Science Foundation of China: No. 31601256.

### Competing Interests

The authors declare that they have no competing interests.

### Author Contributions

- Wennan Su conceived and designed the experiments, performed the experiments, analyzed the data, prepared figures and/or tables, authored or reviewed drafts of the paper, and approved the final draft.
- Shakeel Ahmad analyzed the data, authored or reviewed drafts of the paper, and approved the final draft.
- Irshad Ahmad performed the experiments, prepared figures and/or tables, and approved the final draft.
- Qingfang Han conceived and designed the experiments, analyzed the data, authored or reviewed drafts of the paper, and approved the final draft.

### Data Availability

   The raw measurements are available in the Supplemental File.

## Supplemental Information

Supplemental information for this article can be found online at http://dx.doi.org/10.7717/peerj.10291#supplemental-information.

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
