# Peer review of "Nitrogen fertilization affects maize grain yield through regulating nitrogen uptake, radiation and water use efficiency, photosynthesis and root distribution"

_PeerJ, doi:10.7717/peerj.10291_

## Round 0.1 · original submission · Minor Revisions

This paper is about N and water use. Authors have presented some valuable findings, while more details are needed in several places before it can be accepted. A general introduction to the N fertilizer in the world, not just in Loess Plateau is required. Some details about photosynthetic characteristics and N leaching are also needed. Regarding the differences between years, the potential effect of precipitation should be discussed.

Reviewer 1 ·

Basic reporting

no comment

Experimental design

no comment

Validity of the findings

no comment

Additional comments

Dear editor and authors:

The present manuscript is reporting a study regarding N fertigation affecting maize grain yield by regulating nitrogen uptake, radiation and water use efficiency, photosynthetic and root distribution. The reviewer argue this study with good design and labored works to give some novel information for accessing N application of maize production in the Loess Plateau of China. The overall MS was prepared well. However, a few of suggestions and questions could be considered for revising as follow:

-Title, please change fertigation to fertilization.
-Abstract, please revise the first sentence.
-IN, Line 35-47 as a manuscript for shooting a international journal, I would suggest not using an local issue to start your introduction, you have to focus on an issue worldwide or regional problem in some countries at least. So please rewrite this part.
-MM, Line 115-137 please add refs for these indicators measurements.
-Results and discussion were ok with the description.

Based on above comments, the final recommendation regarding this manuscript is minor revision.

Reviewer 2 ·

Basic reporting

no comment

Experimental design

no comment

Validity of the findings

no comment

Additional comments

This is a well-organized article. I have a little doubt that photosynthetic characteristics may be related to soil moisture and maize varieties. Please add this part to the discussion.

Reviewer 3 ·

Basic reporting

Line 37, although leaching is not a focus in this study, but I feel like it is important to point it out as the excessive N fertilizer not only reducing NUE but causing great leaching (belowground water contamination)

Line 39, omit "application rates vary greatly"? since the point of this sentence is to emphasize the excessive N input.

Line 41, what is approximate level of maize demand?

Line 43, probably also mention about reducing the N leaching, which is a big problem in Ag science.

Line 89, Line 92, "455 m" and "581 mm" are good enough, no need to have one more digit.

Line 94-97, different font size

Line 99, m2, superscript

Line 249-263, please keep consistent about the format of "N0, N150, N225, N300" throughout the manuscript

Figure 1, possibly move to the Appendix since this was a background information, and was not used to explain Results in the Discussion. And only one legend is needed instead of four in four panels.

Figure 2 and 3 can be possibly merged since they are sharing the same experiential settings.

Table 4, last row has five cells displayed in square?

Experimental design

Line 95, were the top 60 cm all organic soils? or having a layer of mineral soils (A horizon)?

Line 160, better to write out the method

Line 163, no need to mention Excel.

Line 163, "presented as SE?" should be "presented with SE"?

Line 164-166, these sentences about ANOVA tests should be moved down to line 195, after calculation of all the index. And the description needs more details, for example, authors also included year as an independent variable in the test, which was not described here.

Validity of the findings

Line 201, were the following sentences saying there was an interaction impact? as the treatment effect was not consistent across years.

Line 228-235, were these statements true in all four years?

Line 243 and line 246, were these two sentences conveying the same idea? omit the second sentence?

Line 265, this first (topic) sentence was too long and included two messages. I suggested starting from "Optimum rate..."

Line 282, reconstruct the sentences. Technically, the N0 is all one type of N application rates.

Line 293, line 301, these two topic sentences read like a Result, please edit them.

What are the reasons of seeing different treatment effects across years? probably due to the variations in precipitation (amount and seasonal distribution) and temperature that shown in Figure 1? Authors should add possible explanation in the Discussion.

Additional comments

This manuscript studied a list of maize index under four levels of N addition over four years. It found that a 25% reduction of current N addition (300 kg ha-1) would still produce an optimal maize growth. Overall this manuscript was well-written but still had some issues as I mentioned above.

---

## Round 0.2 · Minor Revisions

All the questions raised by reviewers have been properly answered and the paper has been obviously improved. However, English needs thorough editing. The authors need to present a language editing certificate when resubmitting the revised paper.

---

## Round 0.3 · accepted · Accept

I am pleased to see the new version with significant changes.